# Berezinskii-Kosterlitz-Thouless transition transport in spin-triplet superconductor

Suk Bum Chung[1,2,3*], Se Kwon Kim[4],

**1** Department of Physics and Natural Science Research Institute, University of Seoul, Seoul 02504, Republic of Korea
**2** School of Physics, Korea Institute for Advanced Study, Seoul 02455, Republic of Korea
**3** Center for Correlated Electron Systems, Institute for Basic Science (IBS), Seoul National University, Seoul 08826, Republic of Korea
**4** Department of Physics, KAIST, Daejeon 34141, Republic of Korea
* sbchung0@uos.ac.kr

July 7, 2021

## Abstract

As the spin-triplet superconductivity arises from the condensation of spinful Cooper pairs, its full characterization requires not only charge ordering, but also spin ordering. For a two-dimensional (2D) easy-plane spin-triplet superconductor, this naïvely seems to suggest the possibility of two distinct Berezinskii-Kosterlitz-Thouless (BKT) phase transitions, one in the charge sector and the other in the spin sector. However, it has been recognized that there are actually three possible BKT transitions, involving the deconfinement of, respectively, the conventional vortices, the merons and the half-quantum vortices with vorticity in both the charge and the spin current. We show how all the transitions can be characterized by the relation between the voltage drop and the spin-polarized current bias. This study reveals that, due to the hitherto unexamined transport of half-quantum vortices, there is an upper bound on the spin supercurrent in a quasi-long range ordered spin-triplet superconductor, which provides a means for half-quantum vortex detection via transport measurements and deeper understanding of fluctuation effects in superconductor-based spintronic devices.

# 1 Introduction

One defining feature of the spin-triplet superconducting phase is the concurrent breaking of spin-rotational symmetry and gauge symmetry [1–4]. Recent years have seen newer candidates for spin-triplet superconductors in uranium-based materials [5–9], doped topological insulator [10, 11], and magic angle graphene [12–17], notwithstanding controversies concerning older candidate $Sr_2RuO_4$ [3, 18, 19]. It has been recognized that the spatial variation of the Cooper pair spin state would lead to spin current carried by Cooper pairs [20–22]. In other words, spin-triplet superconductor can support superfluid spin transport, a spin analogue of mass transport in superfluid [23, 24] in addition to superconducting charge transport.

The multifaceted aspect of spin-triplet superconductivity also gives rise to additional complexity in fluctuations. An especially telling case would be the spin-triplet superconductor in two dimensions (2D) with easy-plane anisotropy, which, as we shall show, has the like-spin pairing when the spin quantization axis is perpendicular. Such two component condensates in 2D would allow multiple types of Berezinskii-Kosterlitz-Thouless (BKT) phase transitions [25, 26] involving both the charge and the spin degrees of freedom. This naturally raises the question of the robustness of superfluid spin transport in 2D spin-triplet superconductors, which is crucial for realizing superconductor-based spintronics with minimal dissipation [27]. Previously, we have shown that quasi-long range ordering is sufficient for spin superfluid transport in the 2D XY magnets [28]. In this Letter, we show that superfluid spin transport of spin-triplet superconductors at finite temperatures fundamentally differs from that of XY magnets due to the existence of fractional vortices, which are topological defects intertwining charge and spin currents. More specifically, we show that the fractional vortex sets an upper bound to spin current, and this upper bound decreases algebraically with distance. Our results indicate the possibility of transport detection of fractional vortices in spin-triplet superconductors. Also, the identified vulnerability of superfluid spin transport to topological defects calls for further investigations of fluctuation effects on promised superconductor-based spintronic devices.

# 2 General considerations

Due to its U(1)×U(1) order parameter, three types of vortices, with therefore three types of BKT transitions in 2D, should be considered for the easy-plane spin-triplet superconductor. This form of order parameter arises because the Cooper pair condensation in the spin-triplet

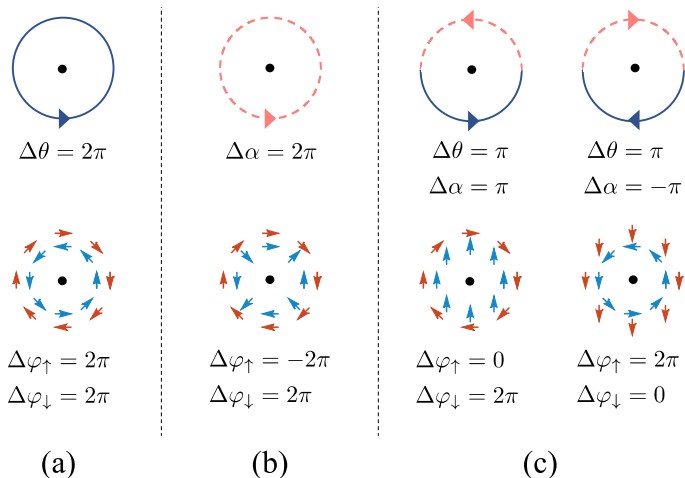

Figure 1: Phase windings of (a) a full-quantum vortex (fqv), (b) a $d$-vector meron (dm), and (c) half-quantum vortices (hqv), where $\theta$ and $\alpha$ represent phases associated with charge and spin order, respectively, and $\varphi_\sigma$ represent the phase of the spin-$\sigma$ Cooper pair $\Delta_{\sigma\sigma} = |\Delta_{\sigma\sigma}|e^{i\varphi_\alpha}$.

superconductor necessarily involves the ordering of the Cooper pair spin state, as shown by the multi-component pairing gap [1–4]

$$i(\mathbf{d} \cdot \boldsymbol{\sigma})\sigma_y = \begin{bmatrix} -d_x + id_y & d_z \\ d_z & d_x + id_y \end{bmatrix} \equiv \begin{bmatrix} \Delta_{\uparrow\uparrow} & \Delta_{\uparrow\downarrow} \\ \Delta_{\downarrow\uparrow} & \Delta_{\downarrow\downarrow} \end{bmatrix}; \tag{1}$$

in the absence of the condensate spin-polarization, the $\mathbf{d}$-vector is real up to an overall phase. For the case where the direction of the $\mathbf{d}$-vector lies in the $xy$-plane, *i.e.* $\hat{\mathbf{d}} = (\cos\alpha, \sin\alpha, 0)$, Eq. (1) shows clearly that only the diagonal elements are non-zero with $\Delta_{\sigma\sigma} = |\Delta|e^{i\pi\frac{1+\sigma}{2}}e^{i(\theta-\sigma\alpha)}$, where $\theta$ is the overall phase. Hence the Cooper pair charge and spin current would be proportional to $\boldsymbol{\nabla}\theta - \frac{2e}{\hbar c}\mathbf{A}$ and $\boldsymbol{\nabla}\alpha$, respectively [22, 29]. As this U(1)×U(1) order parameter remains single-valued for vortices with $m^{\mathrm{c}} \mp m^{\mathrm{sp}} \in \mathbb{Z}$, where $m^{\mathrm{c}}$ ($m^{\mathrm{sp}}$) is the number of $2\pi$ windings of charge (spin) phase $\theta$ ($\alpha$), the basic topological defects supported by the easy-plane spin-triplet superconductor include not only the conventional full-quantum vortex (fqv) with $(m^{\mathrm{c}}, m^{\mathrm{sp}}) = (\pm 1, 0)$ and the $d$-vector meron (dm) with $(m^{\mathrm{c}}, m^{\mathrm{sp}}) = (0, \pm 1)$, but, as illustrated in Fig. 1, also the half-quantum vortex (hqv) with $|m^{\mathrm{c}}| = |m^{\mathrm{sp}}| = 1/2$ [30–33, 45]. It has been shown that fractional vortices can exist for this type of superconducting order parameter even when its physical mechanism is entirely different [34–39]. Given that the BKT transition arises from the (de)confinement of vortices below (above) the transition temperature, the existence of three types of vortices implies the possibility of three distinct types of the BKT transitions in the 2D easy-plane spin-triplet superconductor (see Fig. 2).

These BKT transitions can be conveniently studied by treating vortices as particles [28, 40–44]. For the 2D easy-plane spin-triplet superconductor, its U(1)×U(1) order parameter

allows us to study the energetics of quenched vortices from the Coulomb gas action [45, 46] [1]

$$
\begin{aligned}
S_{\mathrm{eff}} = {} & 2\pi\hbar \sum_i \int^{\mathbf{r}_i} d\mathbf{r}' \cdot \left[ \frac{m_i^{\mathrm{c}}}{2e} \mathbf{J}_{\mathrm{c}}^{\mathrm{ext}}(\mathbf{r}') - \frac{m_i^{\mathrm{sp}}}{\hbar} \mathbf{J}_{\mathrm{sp}}^{z,\mathrm{ext}}(\mathbf{r}') \right] \times \hat{\mathbf{z}} \\
& - 2\pi \sum_{i \neq j} \left( m_i^{\mathrm{c}} m_j^{\mathrm{c}} K_{\mathrm{c}} + m_i^{\mathrm{sp}} m_j^{\mathrm{sp}} K_{\mathrm{sp}} \right) \log \frac{|\mathbf{r}_i - \mathbf{r}_j|}{\xi},
\end{aligned} \tag{2}
$$

where $\mathbf{J}_{\mathrm{c}}^{\mathrm{ext}}$ ($\mathbf{J}_{\mathrm{sp}}^{z,\mathrm{ext}}$) is the externally applied charge (spin) current, while the second term is the vortex-vortex interaction energy with $\xi$ being the vortex core radius and $K_{\mathrm{c}}(K_{\mathrm{sp}})$ the phase stiffness for charge (spin). From the second term fo Eq. (2), the BKT temperature for each vortex type can be determined by adopting the known results for superfluids and superconductors [36, 46, 49–51]:

$$
T_{\mathrm{BKT}}(|m^{\mathrm{c}}|, |m^{\mathrm{sp}}|) = \frac{\pi}{2k_B} \left[ (m^{\mathrm{c}})^2 K_{\mathrm{c}} + (m^{\mathrm{sp}})^2 K_{\mathrm{sp}} \right].
$$

Upon increasing the temperature, among the three possible BKT transitions, the one with the lowest critical temperature would disorder the spin-triplet superconductivity, and it is determined by, as shown in Fig. 2, the stiffness ratio $K_{\mathrm{sp}}/K_{\mathrm{c}}$: $K_{\mathrm{sp}}/K_{\mathrm{c}} > 3$, $K_{\mathrm{sp}}/K_{\mathrm{c}} < 1/3$, and $1/3 < K_{\mathrm{sp}}/K_{\mathrm{c}} < 3$ correspond to the fqv, the dm, and the hqv deconfinement, respectively [2]. It is possible to derive from Eq. (2), together with phenomenological vortex mobility that we shall introduce below, the DC transport change at the BKT transitions from the presence (absence) of the free vorticity density $n_{\mathrm{c/sp}}^f$ above (below) the transition.

Specifically, above the transition temperature, where the free vortex density is finite, by assuming the vortex mobility to be purely longitudinal to the force from applied current for simplicity, we obtain the following relation between the applied current and the vortex current density:

$$
\mathbf{j}(m^{\mathrm{c}}, m^{\mathrm{sp}}) = 2\pi\hbar\mu n^f \left[ \frac{\mathrm{sgn}(m^{\mathrm{c}})}{2e} \mathbf{J}_{\mathrm{c}}^{\mathrm{ext}} - \frac{\mathrm{sgn}(m^{\mathrm{sp}})}{\hbar} \mathbf{J}_{\mathrm{sp}}^{z,\mathrm{ext}} \right] \times \hat{\mathbf{z}} \tag{3}
$$

where $\mu$ is the vortex mobility and $n^f$ is the free vortex density; we take $\mathrm{sgn}(0) = 0$. The contribution of this vortex current to the charge/spin vorticity current is $\mathbf{j}_{\mathrm{c/sp}} = m^{\mathrm{c/sp}} \mathbf{j}(m^{\mathrm{c}}, m^{\mathrm{sp}})$. Note that the linear relationship between the charge/spin current and the charge/spin vorticity current holds above the transition temperature.

Below the BKT transition, where free vortices are absent, the energy barrier against the unbinding of bound vortex pairs is finite in the presence of an external current density, with the energy barrier per vortex of

$$
\Delta E \approx 2k_B T_{\mathrm{BKT}}(|m^{\mathrm{c}}|, |m^{\mathrm{sp}}|) \log \frac{2k_B T_{\mathrm{BKT}}(|m^{\mathrm{c}}|, |m^{\mathrm{sp}}|)}{\pi\hbar\xi \left| \frac{m^{\mathrm{c}}}{2e} J_{\mathrm{c}}^{\mathrm{ext}} - \frac{m^{\mathrm{sp}}}{\hbar} J_{\mathrm{sp}}^{z,\mathrm{ext}} \right|} \tag{4}
$$

for sufficiently small charge and spin currents, which implies thermal dissociation probability $\propto \exp(-\Delta E/k_B T)$ for bound vortex pairs. Therefore, the current $\mathbf{j}(m^{\mathrm{c}}, m^{\mathrm{sp}})$ is linear in applied current above the transition but nonlinear with the exponent of $1 + 2T_{\mathrm{BKT}}/T$ below the

---

[1]In chiral superconductor, additional topological terms may be present, giving rise to the non-Abelian hqv braid statistics [47] plus a universal Abelian vortex exchange phase factor [48], neither of which, however, contribute to energetics of the vortex pair unbinding.

[2]Possible complications from thermal fluctuation of more than two vortex types are ignored in this work.

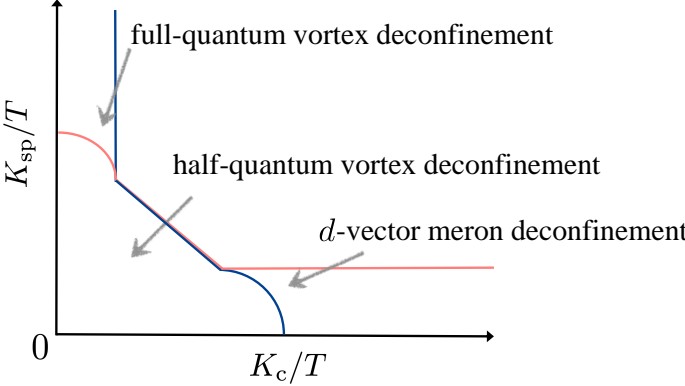

Figure 2: Three distinct Bereziskii-Kosterlitz-Thouless transitions driven by deconfinements of three different types vortices in the $(K_{\mathrm{sp}}/T, K_{\mathrm{c}}/T)$ plane, where the arrow directions correspond to increasing temperature and the blue (red) curve indicates phase transition in the charge (spin) sector. Deconfinements of full-quantum vortex and $d$-vector meron leads to the exponentially decaying phase coherence of charge order and spin order, respectively, whereas deconfinement of half-quantum vortex the exponentially decaying phase coherence of both orders.

transition [28, 49, 50], and the transport signature of the BKT transition consists of measuring this change of vortex current.

Yet, due to the finite spin lifetime, the effect of a vortex current on transport properties differs qualitatively between the charge and the spin sector. First, for the charge sector, charge conservation dictates that incoming and outgoing current should be equal. Measuring the current-voltage relation should give us the charge vorticity current through the Josephson relation,

$$\mathbf{E}_J = 2\pi \frac{\hbar}{2e}\hat{\mathbf{z}} \times \mathbf{j}_{\mathrm{c}} = \frac{\pi\hbar}{e}\hat{\mathbf{z}} \times \sum_{m^{\mathrm{c}}, m^{\mathrm{sp}}} m^{\mathrm{c}}\mathbf{j}(m^{\mathrm{c}}, m^{\mathrm{sp}}). \tag{5}$$

indicating that the qualitative change of current-voltage relation across the BKT transition represents the corresponding change of the charge vorticity current. Second, for the spin sector, the spin analogue of the Josephson relation Eq. (5) [28, 44, 52, 53] gives us $\boldsymbol{\nabla}s_z = 2\pi(K_{\mathrm{sp}}/v_{\mathrm{sp}}^2)\hat{\mathbf{z}} \times \mathbf{j}_{\mathrm{sp}}$, where $s_z$ is the spin density of Cooper pairs and $v_{\mathrm{sp}}$ is the spin-mode velocity. However, while charge conservation ensures uniform DC charge current, the same does not hold for spin due to the finite lifetime of spin. Indeed, $\boldsymbol{\nabla} \cdot \mathbf{J}_{\mathrm{sp}}^z + s_z/\tau = 0$ for DC transport [28, 29] with $\tau$ spin lifetime leads to

$$\boldsymbol{\nabla}(\boldsymbol{\nabla} \cdot \mathbf{J}_{\mathrm{sp}}) = 2\pi \frac{K_{\mathrm{sp}}}{\tau v_{\mathrm{sp}}^2}\mathbf{j}_{\mathrm{sp}} \times \hat{\mathbf{z}}. \tag{6}$$

This implies that the change in the spin vorticity current $\mathbf{j}_{\mathrm{sp}}$ across the BKT transition can be detected by the change in the spatial profile of the spin current $\mathbf{J}_{\mathrm{sp}}$, which, as we shall show below, manifests through the dependence of magnetoresistance on the distance between the source and drain of spin current.

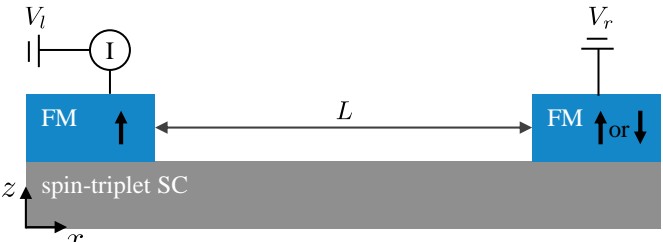

Figure 3: Illustration of an experimental setup for transport experiment, where a spin-triplet superconductor (SC) is used as a charge- and spin-transport medium and two ferromagnetic metals (FMs) serve as magnetic leads with distance $L$.

## 3 Transport signature of BKT transitions

### 3.1 Transport setup

The distinct signature of each BKT transition can be detected from the DC current-voltage relation for the setup of Fig. 3, which has been used in Ref. [22] to study zero-temperature magnetoresistance of spin-triplet superconductors. It consists of two leads made of ferromagnetic metal (*e.g.* $SrRuO_3$) attached to the spin-triplet superconductor, with the lead magnetization perpendicular to the superconductor **d**-vector. One possible material candidate is the recently fabricated $SrRuO_3|Sr_2RuO_4$ heterostructure [54]. The boundary condition for the current of spin-$\sigma$ Cooper pairs is

$$I_{l,r}^{\sigma} = \pm g_{l,r}^{\sigma\sigma} \left( V_{l,r} - \frac{\hbar}{2e} \partial_t \varphi_\sigma^{l,r} \right), \tag{7}$$

where $g_{l,r}^{\sigma\sigma}$ is the spin-$\sigma$ conductance for the left/right lead, $V_{l,r}$ is the voltage on the normal side of the left/right lead, and $\varphi_\sigma \equiv \arg \Delta_{\sigma\sigma}$. The corresponding boundary condition for the spin current is

$$I_{l,r}^{\mathrm{sp}} = \frac{\hbar}{2e} p_{l,r} I \pm (1 - p_{l,r}^2) \tilde{g}_{l,r} \frac{d}{dx} I_{l,r}^{\mathrm{sp}}, \tag{8}$$

where $\tilde{g}_{l,r} \equiv \frac{\hbar^2}{4e^2} \frac{K_{\mathrm{sp}}}{v_{\mathrm{sp}}^2} \frac{\tau}{w} g_{l,r}$ ($w$ is the lead width and $g_{l,r} \equiv \sum_\sigma g_{l,r}^{\sigma\sigma}$). What will be measured in Fig. 3 is the bulk current-voltage relation,

$$\begin{aligned} \Delta V &\equiv (V_l - V_r) - \frac{g_l + g_r}{g_l g_r} I \\ &= \frac{\hbar}{2e} (\partial_t \theta_l - \partial_t \theta_r) - \frac{\hbar}{2e} \frac{K_{\mathrm{sp}}}{v_{\mathrm{sp}}^2} \frac{\tau}{w} \left( p_l \frac{d}{dx} I_l^{\mathrm{sp}} - p_r \frac{d}{dx} I_r^{\mathrm{sp}} \right), \end{aligned} \tag{9}$$

where $p_{l,r} \equiv \sum_\sigma \sigma g_{l,r}^{\sigma\sigma} / g_{l,r}$ is the contact conductance spin polarization; we have also used $\partial_t \alpha_{l,r} = -\frac{K_{\mathrm{sp}}}{v_{\mathrm{sp}}^2} s_z^{l,r} = \frac{K_{\mathrm{sp}}}{v_{\mathrm{sp}}^2} \frac{\tau}{w} \frac{d}{dx} I_{l,r}^{\mathrm{sp}}$, derived from the finite spin lifetime and $[s_z(\mathbf{r}), \alpha(\mathbf{r}')] = i\delta(\mathbf{r} - \mathbf{r}')$. In addition, as shown in Fig. 3, the current is constrained to flow along the $x$-direction, with $x = \mp L/2$ for the left / right lead. The novelty in our BKT transport arises from the second term of Eq. (9) that gives the magnetoresistance through $p_{l,r}$, which, as we shall show, allows for electrical detection of the spin vorticity deconfinement.

## 3.2 Full-quantum vortex deconfinement

While the fqv deconfinement, as it consists of the charge vorticity deconfinement alone, is of the same type as the BKT transition of conventional superconductors with respect to charge transport [49, 50, 55–57], spin transport reveals the fqv deconfined phase to be unconventional. In the case of the fqv deconfinement, the change in the first term of $\Delta V$ in Eq. (9), $(\partial_t \theta_l - \partial_t \theta_r)/2\pi = \int_{-L/2}^{+L/2} dx\, j_c$, through the transition is indeed exactly that of the BKT transition in conventional superconductor. However, the voltage-current relation across the fqv deconfinement is modified by the second term in Eq. (9) involving the spin current $I^{\mathrm{sp}}$, which cannot be absent in the Fig. 3 setup.

In the considered setup, due to the absence of spin vorticity both above and below the transition, the spin torque always vanishes in the bulk, leading to a uniform $s_z$, and hence a spin current linear in the distance $x$ from the leads [22, 29]. The linear profile of the spin current, together with Eq. (8) gives us

$$\frac{\frac{\hbar}{2e}(p_l - p_r)I}{dI^{\mathrm{sp}}/dx} = -L - [(1 - p_l^2)\tilde{g}_l + (1 - p_r^2)\tilde{g}_r],$$

and inserting this into Eq. (9) gives us the following current-voltage relation in the large-$L$ limit,

$$\Delta V = \frac{\hbar}{2e}\frac{K_{\mathrm{sp}}}{v_{\mathrm{sp}}^2}\frac{\tau}{w}\frac{(p_l - p_r)^2}{L}I$$
$$+ \begin{cases} \frac{L}{w}\rho_0^{\mathrm{fqv}}I\left|\frac{I}{I_0^{\mathrm{fqv}}}\right|^{2T_{\mathrm{BKT}}^{\mathrm{fqv}}/T} & (T < T_{\mathrm{BKT}}^{\mathrm{fqv}}), \\ \frac{L}{w}\left(\frac{\pi\hbar}{e}\right)^2 \mu_{\mathrm{fqv}} n_f I & (T > T_{\mathrm{BKT}}^{\mathrm{fqv}}), \end{cases} \tag{10}$$

where $\rho_0^{\mathrm{fqv}}$ and $I_0^{\mathrm{fqv}}$ are phenomenological constants in the units of 2D resistivity and current, respectively [3]; this is our first main result. Below or above $T_{\mathrm{BKT}}^{\mathrm{fqv}}$, the current-voltage relation in absence of the spin polarization is exactly that of the conventional BKT transition. The difference lies in magnetoresistance $V_{\mathrm{mr}} \equiv \Delta V(p_l = -p_r = p) - \Delta V(p_l = p_r = p)$, which remains unchanged across the BKT transition and reproduces the result of Ref. [22]: $V_{\mathrm{mr}} \propto I/L$. This implies that, as indicated in Fig. 1, we have an unconventional phase above the transition that, while not superconducting, retains some of spin quasi-ordering from the spin-triplet superconductivity. This is because, when only the overall phase is disordered, there still remains spin nematicity, defined from the order parameter of Eq. (1) as quasi-long range ordered $\Delta_{\uparrow\uparrow}\Delta_{\downarrow\downarrow}^* \propto (\hat{d}_x^2 - \hat{d}_y^2) - i2\hat{d}_x\hat{d}_y$.

## 3.3 $d$-vector meron deconfinement

Conversely, the deconfinement of spin vortices, i.e. dm, directly disturbs spin transport (as merons interrupt spin transport in 2D XY magnets [28]), giving rise to the change in the spin transport equation, i.e.

$$\frac{I_0^{\mathrm{sp}}}{I^{\mathrm{sp}}}\frac{d^2}{dx^2}\frac{I^{\mathrm{sp}}}{I_0^{\mathrm{sp}}} = \begin{cases} \lambda^{-2}\left|\frac{I^{\mathrm{sp}}}{I_0^{\mathrm{sp}}}\right|^{2T_{\mathrm{BKT}}^{\mathrm{dm}}/T} & (T < T_{\mathrm{BKT}}^{\mathrm{dm}}), \\ \lambda_0^{-2} & (T > T_{\mathrm{BKT}}^{\mathrm{dm}}), \end{cases} \tag{11}$$

---

[3]Vortices with the opposite winding may not have the same mobility in the chiral superconductor; in that case $\mu_{\mathrm{fqv}}$ should be interpreted as the average mobility.

where $I_0^{\mathrm{sp}}$ and $\lambda$ are phenomenological constants in the units of spin current and length, respectively, and $\lambda_0 = \sqrt{\frac{v_{\mathrm{sp}}^2 \tau}{\mu_{\mathrm{dm}} n_f}}$ is the spin diffusion length derived from Eqs. (3) and (6) [28]. Meanwhile the absence of the charge vorticity current means $(\partial_t \theta_l - \partial_t \theta_r)/2\pi = \int_{-L/2}^{+L/2} dx \, j_c = 0$ holds across the transition. Hence, from Eq. (9), we can see that this transition would manifest in the current-voltage relation only through the change in magnetoresistance. Although Eq. (11) below the transition is nonlinear, our analysis is facilitated by its being readily reducible to a first-order differential equation for the symmetric (antisymmetric) lead configuration, $p_l = \pm p_r$ with $g_l = g_r = g$.

By solving the bulk equations of motion in conjunction with charge and spin boundary conditions in the large-$L$ limit, and keeping only the leading-order functional dependence on $I$ and $L$, we obtain the following magnetoresistance:

$$V_{\mathrm{mr}} \propto \left( \frac{L}{\lambda} \right)^{-2 - \frac{2T}{T_{\mathrm{BKT}}^{\mathrm{dm}}}} \left| \frac{\hbar I}{2 e I_0^{\mathrm{sp}}} \right|^{-1 - \frac{T_{\mathrm{BKT}}^{\mathrm{dm}}}{T}} , \qquad (12)$$

for $T \lesssim T_{\mathrm{BKT}}^{\mathrm{dm}}$ in the $L \gg \lambda |e I_0^{\mathrm{sp}} / p \hbar I|^{\frac{T_{\mathrm{BKT}}^{\mathrm{dm}}}{T}}$ limit and

$$V_{\mathrm{mr}} \propto I e^{-L/\lambda_0} , \qquad (13)$$

for $T > T_{\mathrm{BKT}}^{\mathrm{dm}}$, which constitute our second main result. It indicates that for $T < T_{\mathrm{BKT}}^{\mathrm{dm}}$ the magnetoresistance decreases algebraically with $L$ as $V_{\mathrm{mr}} \sim L^{-2 - 2T/T_{\mathrm{BKT}}^{\mathrm{dm}}}$, but vanishes exponentially with $L$ for $T > T_{\mathrm{BKT}}^{\mathrm{dm}}$ with the decay length given by the spin diffusion length $\lambda_0$ defined above. This is qualitatively equivalent to the change of spin transport across the magnetic BKT transition in easy-plane magnets [28]. The complete expressions, as shown in Appendix A, also gives us the vanishing of all dissipation in absence of any spin-polarization, i.e. $\Delta V(p_l = p_r = 0) = 0$, on both sides of the transition. This can be taken as the evidence for the persistence of superconductivity, i.e. absence of dissipation from the charge degree of freedom, above the dm BKT transition where spin ordering is destroyed, as indicated in Fig. 1. This indicates a charge-4$e$ superconductivity involving pairing of two Cooper pairs, i.e. $\Delta_{\uparrow\uparrow} \Delta_{\downarrow\downarrow} \propto e^{i2\theta}$ [12, 36].

## 3.4 Half-quantum vortex deconfinement

As an hqv possesses vorticity in both charge and spin, separation of the charge and the spin transport is not guaranteed in presence of the hqv current. This can be revealed only through spin-polarized current bias; otherwise transport effects of hqv's will not differ qualitatively from those of fqv's, as one can infer from Eq. (2). To treat this, it needs to be noted that two types of hqv's, each with $m^{\mathrm{c}} - \sigma m^{\mathrm{sp}} = \pm 1$ and therefore, as shown in Fig. 1 (c), a nonzero vorticity only for $\Delta_{\sigma\sigma}$, both have deconfinement onset at $T = T_{\mathrm{BKT}}^{\mathrm{hqv}}$ in absence of bulk spin-polarization, but their contributions to the spin vorticity current have opposite signs. Hence, for $T < T_{\mathrm{BKT}}^{\mathrm{hqv}}$,

$$j_{\mathrm{sp}} = \frac{\hbar v_{\mathrm{sp}}^2 \tau}{4 \pi K_{\mathrm{sp}} w \tilde{\lambda}^2} \sum_{\sigma} \sigma \frac{I + \sigma \frac{2e}{\hbar} I^{\mathrm{sp}}}{2e} \left| \frac{I + \sigma \frac{2e}{\hbar} I^{\mathrm{sp}}}{I_0^{\mathrm{hqv}}} \right|^{\frac{2 T_{\mathrm{BKT}}^{\mathrm{hqv}}}{T}} , \qquad (14)$$

where $I_0^{\mathrm{hqv}}$ and $\tilde{\lambda}$ are phenomenological constants in the unit of current and length. Combined with Eq. (6), this leads to the spin current differential equation

$$\frac{d^2}{dx^2}\frac{I^{\mathrm{sp}}}{\frac{\hbar}{2e}I_0^{\mathrm{hqv}}} \approx \frac{1+\frac{2T_{\mathrm{BKT}}^{\mathrm{hqv}}}{T}}{\tilde{\lambda}^2}\left|\frac{I}{I_0^{\mathrm{hqv}}}\right|^{\frac{2T_{\mathrm{BKT}}^{\mathrm{hqv}}}{T}}\frac{I^{\mathrm{sp}}}{\frac{\hbar}{2e}I_0^{\mathrm{hqv}}}. \tag{15}$$

for most of the bulk in the large-$L$ limit, as the finite spin lifetime leads to $I^{\mathrm{sp}} \ll \frac{\hbar}{2e}I$. For the same reason, the charge vorticity current in this limit can be taken to be nearly independent of the spin current, $j_c \propto |I|^{1+2T_{\mathrm{BKT}}^{\mathrm{hqv}}/T}$. This indicates that, while the bound hqv's behave like the bound fqv's in the charge sector, they induce, in contrast to the bound dm's, the exponential decay of the spin current when the charge current has a finite magnitude, with the spin decay length of

$$\tilde{\lambda}_{\mathrm{eff}} \equiv \frac{\tilde{\lambda}}{\sqrt{1+2T_{\mathrm{BKT}}^{\mathrm{hqv}}/T}}\left|\frac{I_0^{\mathrm{hqv}}}{I}\right|^{\frac{T_{\mathrm{BKT}}^{\mathrm{hqv}}}{T}}.$$

For this case, we have the magnetoresistance voltage:

$$V_{\mathrm{mr}} \sim I e^{-L/\tilde{\lambda}_{\mathrm{eff}}} \quad \text{for } T < T_{\mathrm{BKT}}^{\mathrm{hqv}}, \tag{16}$$

which decays exponentially as a function of the lead spacing $L$ with the length scale given by $\tilde{\lambda}_{\mathrm{eff}}$. Since the decaying length $\tilde{\lambda}_{\mathrm{eff}}$ diverges as $I \to 0$, the long-range spin transport do survive albeit at the price of setting the maximum spin current $I_{\max}^{\mathrm{sp}} \sim p_l(\tilde{\lambda}/L)^{T/T_{\mathrm{BKT}}^{\mathrm{hqv}}}\frac{\hbar}{2e}I_0^{\mathrm{hqv}}$ at $T > 0$ for the given lead spacing $L$.

For $T > T_{\mathrm{BKT}}^{\mathrm{hqv}}$, by contrast, it is possible to separate the charge and the spin sector, *i.e.* the charge (spin) vorticity current is proportional to the charge (spin) current and independent of the spin (charge) current as the densities of all hqv types are equal and current independent to the leading order. Therefore, above the transition, we have

$$V_{\mathrm{mr}} \propto I e^{-L/\tilde{\lambda}_0} \quad \text{for } T > T_{\mathrm{BKT}}^{\mathrm{hqv}}, \tag{17}$$

where $\tilde{\lambda}_0^2 \equiv \frac{v_{\mathrm{sp}}^2\tau}{\mu_{\mathrm{hqv}}n_f}$. The magnetoresistance voltage decays exponentially as a function of $L$ akin to the low-temperature case, but the decaying length $\tilde{\lambda}_0$ is finite in the limit of $I \to 0$. Eqs. (16), (17) are our third main results. For the full expressions of $V_{\mathrm{mr}}$ for the hqv case, see Appendix B.

## 4 Conclusion

We have explained how the hqv-driven BKT transition displays transport characteristics absent in the fqv-driven or dm-driven BKT transitions. Experimental detection for fixed $L$ can be made by examining how $V_{\mathrm{mr}}/I$ ratio depends on $I$. For $T > T_{\mathrm{BKT}}^{\mathrm{hqv}}$, $L > \tilde{\lambda}_0$ ensures exponentially vanishing $V_{\mathrm{mr}}/I$ independent of $I$. For $T < T_{\mathrm{BKT}}^{\mathrm{hqv}}$, $V_{\mathrm{mr}}/I$ will not be exponentially small for $I$ smaller than the critical value that scales as $L^{-T/T_{\mathrm{BKT}}^{\mathrm{hqv}}}$.

We expect our results to be applicable to many of superconductors where U(1)×U(1) order parameters arise from various mechanisms [34–39] All these superconductors can be

regarded as having a two component condensates, and our hqv transport results of Eqs. (16) and (17) should hold if in-plane anisotropy is very weak or arising from the hexagonal crystal field [58] (a frequent feature in van der Waals materials) and any imbalance between the two components has a finite relaxation time.

## Acknowledgements

**Funding information**   We would like to thank Yaroslav Tserkovnyak, Steve Kivelson, Eduardo Fradkin, and S. Raghu and for sharing their insights into the BKT physics, and Tae Won Noh for motivating this work. S.B.C. was supported by the National Research Foundation of Korea (NRF) grants funded by the Korea government (MSIT) (2020R1A2C1007554) and the Ministry of Education (2018R1A6A1A06024977). S.K.K. was supported by Brain Pool Plus Program through the National Research Foundation of Korea (NRF) funded by the Ministry of Science and ICT (2020H1D3A2A03099291), by the National Research Foundation of Korea (NRF) grant funded by the Korea government (MSIT) (2021R1C1C1006273), and by the National Research Foundation of Korea funded by the Korea Government via the SRC Center for Quantum Coherence in Condensed Matter (2016R1A5A1008184).

## A   $\Delta V$ near the $d$-vector meron deconfinement

Solving Eq. (11) for $T < T_{\mathrm{BKT}}^{\mathrm{dm}}$ with the boundary condition of Eq. (8) becomes simpler for the symmetric / antisymmetric setup $p_l = \pm p_r = p$ and $g_l = g_r \equiv g$ as it will give us the spin current $I^{\mathrm{sp}}$ that is an even / odd function of $x$. This enables us to obtain a first order differential equations for the spin current:

$$\lambda \frac{d}{dx} \frac{I^{\mathrm{sp}}}{I_0^{\mathrm{sp}}} = \sqrt{\frac{\left| \frac{I^{\mathrm{sp}}}{I_0^{\mathrm{sp}}} \right|^{2+2T_{\mathrm{BKT}}^{\mathrm{dm}}/T} - \left| \frac{I^{\mathrm{sp}}(x=0)}{I_0^{\mathrm{sp}}} \right|^{2+2T_{\mathrm{BKT}}^{\mathrm{dm}}/T}}{1 + T_{\mathrm{BKT}}^{\mathrm{dm}}/T}} \tag{18}$$

for the symmetric setup at $x > 0$ and

$$\lambda \frac{d}{dx} \frac{I^{\mathrm{sp}}}{I_0^{\mathrm{sp}}} = -\sqrt{\frac{\left| \frac{I^{\mathrm{sp}}}{I_0^{\mathrm{sp}}} \right|^{2+2T_{\mathrm{BKT}}^{\mathrm{dm}}/T}}{1 + T_{\mathrm{BKT}}^{\mathrm{dm}}/T} + \left[ \lambda \frac{d}{dx}\bigg|_{x=0} \frac{I^{\mathrm{sp}}(x)}{I_0^{\mathrm{sp}}} \right]^2} \tag{19}$$

for the antisymmetric setup at $x > 0$.

For the symmetric setup below the transition, we obtain

$$\begin{aligned}
\frac{d}{dx} I_r^{\mathrm{sp}} = &\frac{1}{\lambda} \frac{\hbar}{2e} pI \frac{1}{\sqrt{1 + T_{\mathrm{BKT}}^{\mathrm{dm}}/T}} \left| \frac{p\hbar I}{2e I_0^{\mathrm{sp}}} \right|^{T_{\mathrm{BKT}}^{\mathrm{dm}}/T} \\
&\times \sqrt{1 - C_{even}(T) \left( \frac{L/2\lambda}{\sqrt{1 + \frac{T_{\mathrm{BKT}}^{\mathrm{dm}}}{T}}} \left| \frac{p\hbar I}{2e I_0^{\mathrm{sp}}} \right|^{\frac{T_{\mathrm{BKT}}^{\mathrm{dm}}}{T}} + \frac{T}{T_{\mathrm{BKT}}^{\mathrm{dm}}} \right)^{-2 - \frac{2T}{T_{\mathrm{BKT}}^{\mathrm{dm}}}}} ,
\end{aligned} \tag{20}$$

where $C_{even}(T) = \left[\sqrt{\pi}\dfrac{T}{T_{\mathrm{BKT}}^{\mathrm{dm}}}\dfrac{\Gamma\left(\frac{3+2T/T_{\mathrm{BKT}}^{\mathrm{dm}}}{2+2T/T_{\mathrm{BKT}}^{\mathrm{dm}}}\right)}{\Gamma\left(\frac{2+T/T_{\mathrm{BKT}}^{\mathrm{dm}}}{2+2T/T_{\mathrm{BKT}}^{\mathrm{dm}}}\right)}\right]^{2+2T/T_{\mathrm{BKT}}^{\mathrm{dm}}}$, by noting that (8) gives us $I_r^{\mathrm{sp}} \approx p\hbar I/2e$ for $T > 0$ in the $I \ll 2eI_0^{\mathrm{sp}}/\hbar$ limit and inserting the relation between $I^{\mathrm{sp}}(x = 0)$ and $I_r^{\mathrm{sp}}$ obtained from,

$$X = \int_1^Y \frac{dy}{\sqrt{y^\alpha - 1}} = \int_1^\infty \frac{dy}{\sqrt{y^\alpha - 1}} - \int_X^\infty y^{-\frac{\alpha}{2}}\left(1 + \frac{1}{2}y^{-\alpha} + \cdots\right)dy$$
$$= \frac{\sqrt{\pi}}{\frac{\alpha}{2} - 1}\frac{\Gamma(3/2 - 1/\alpha)}{\Gamma(1 - 1/\alpha)} - \frac{1}{\left(\frac{\alpha}{2} - 1\right)Y^{\frac{\alpha}{2}-1}} + O(Y^{-\frac{3\alpha}{2}+1})$$

back into Eq. (18). This gives us

$$\Delta V_{even} = 2\frac{p}{g}\frac{2e}{\hbar}\tilde{g}\frac{d}{dx}I_r^{\mathrm{sp}}$$

$$= p^2\frac{2I}{g}\frac{\tilde{g}/\lambda}{\sqrt{1 + \frac{T_{\mathrm{BKT}}^{\mathrm{dm}}}{T}}}\left|\frac{p\hbar I}{2eI_0^{\mathrm{sp}}}\right|^{\frac{T_{\mathrm{BKT}}^{\mathrm{dm}}}{T}}\left[1 - \frac{1}{2}C_{even}(T)\left(\frac{L/2\lambda}{\sqrt{1 + \frac{T_{\mathrm{BKT}}^{\mathrm{dm}}}{T}}}\left|\frac{p\hbar I}{2eI_0^{\mathrm{sp}}}\right|^{\frac{T_{\mathrm{BKT}}^{\mathrm{dm}}}{T}}\right)^{-2 - \frac{2T}{T_{\mathrm{BKT}}^{\mathrm{dm}}}}\right]$$

$$(21)$$

in the limit taken for Eq. (12).

For the antisymmetric setup below the transition, we obtain

$$\frac{d}{dx}I_r^{\mathrm{sp}} = -\frac{1}{\lambda}\frac{\hbar}{2e}pI\frac{1}{\sqrt{1 + T_{\mathrm{BKT}}^{\mathrm{dm}}/T}}\left|\frac{p\hbar I}{2eI_0^{\mathrm{sp}}}\right|^{T_{\mathrm{BKT}}^{\mathrm{dm}}/T}$$

$$\times \sqrt{1 + C_{odd}(T)\left[\frac{L/2\lambda}{\sqrt{1 + \frac{T_{\mathrm{BKT}}^{\mathrm{dm}}}{T}}}\left|\frac{p\hbar I}{2eI_0^{\mathrm{sp}}}\right|^{\frac{T_{\mathrm{BKT}}^{\mathrm{dm}}}{T}} + \frac{T}{T_{\mathrm{BKT}}^{\mathrm{dm}}}\right]^{-2 - \frac{2T}{T_{\mathrm{BKT}}^{\mathrm{dm}}}}},$$

$$(22)$$

where $C_{odd}(T) = \left[\dfrac{\Gamma\left(\frac{T_{\mathrm{BKT}}^{\mathrm{dm}}/T}{2+2T_{\mathrm{BKT}}^{\mathrm{dm}}/T}\right)\Gamma\left(\frac{3+2T_{\mathrm{BKT}}^{\mathrm{dm}}/T}{2+2T_{\mathrm{BKT}}^{\mathrm{dm}}/T}\right)}{\sqrt{\pi(1+T_{\mathrm{BKT}}^{\mathrm{dm}}/T)}}\right]^2$, by noting that (8) gives us $I_r^{\mathrm{sp}} \approx -p\hbar I/2e$ for $T > 0$ in the $I \ll 2eI_0^{\mathrm{sp}}/\hbar$ limit and inserting the relation between $I^{\mathrm{sp}}(x = 0)$ and $I_r^{\mathrm{sp}}$ obtained from,

$$X = \int_0^Y \frac{dy}{\sqrt{y^\alpha + 1}} = \int_0^\infty \frac{dy}{\sqrt{y^\alpha + 1}} - \int_Y^\infty y^{-\frac{\alpha}{2}}\left(1 - \frac{1}{2}y^{-\alpha} + \cdots\right)dy$$
$$= \frac{\Gamma(1/2 - 1/\alpha)\Gamma(1 + 1/\alpha)}{\sqrt{\pi}} - \frac{1}{\left(\frac{\alpha}{2} - 1\right)Y^{\frac{\alpha}{2}-1}} + O(Y^{-\frac{3\alpha}{2}+1})$$

back into Eq. (19). This gives us

$$\Delta V_{odd} = -2\frac{p}{g}\frac{2e}{\hbar}\tilde{g}\frac{d}{dx}I_r^{\mathrm{sp}}$$

$$= p^2\frac{2I}{g}\frac{\tilde{g}/\lambda}{\sqrt{1+\frac{T_{\mathrm{BKT}}^{\mathrm{dm}}}{T}}}\left|\frac{p\hbar I}{2eI_0^{\mathrm{sp}}}\right|^{\frac{T_{\mathrm{BKT}}^{\mathrm{dm}}}{T}}\left[1+\frac{1}{2}C_{odd}(T)\left(\frac{L/2\lambda}{\sqrt{1+\frac{T_{\mathrm{BKT}}^{\mathrm{dm}}}{T}}}\left|\frac{p\hbar I}{2eI_0^{\mathrm{sp}}}\right|^{\frac{T_{\mathrm{BKT}}^{\mathrm{dm}}}{T}}\right)^{-2-\frac{2T}{T_{\mathrm{BKT}}^{\mathrm{dm}}}}\right]$$

(23)

in the limit taken for Eq. (12). From Eqs. (21) and (23)

$$V_{\mathrm{mr}} = \Delta V_{even} - \Delta V_{odd}$$

gives us the magnetoresistance of Eq. (12).

For $T > T_{\mathrm{BKT}}^{\mathrm{dm}}$, the general solution of the now-linear Eq. (11),

$$I^{\mathrm{sp}}(x) = \frac{1}{2}(I_l^{\mathrm{sp}} + I_r^{\mathrm{sp}})\frac{\cosh(x/\lambda_0)}{\cosh(L/2\lambda_0)} - \frac{1}{2}(I_l^{\mathrm{sp}} - I_r^{\mathrm{sp}})\frac{\sinh(x/\lambda_0)}{\sinh(L/2\lambda_0)},$$

can be inserted into Eq. (8) to give us, up to the first order in $e^{-L/\lambda_0}$,

$$\frac{d}{dx}I_l^{\mathrm{sp}} = -\frac{1}{\lambda_0}(I_l^{\mathrm{sp}} - 2I_r^{\mathrm{sp}}e^{-L/\lambda_0}) = -\frac{\hbar}{2e}\frac{I}{\lambda_0}\frac{p_l\left[1+(1-p_r^2)\frac{\tilde{g}_r}{\lambda_0}\right] - 2p_re^{-L/\lambda_0}}{\left[1+(1-p_l^2)\frac{\tilde{g}_l}{\lambda_0}\right]\left[1+(1-p_r^2)\frac{\tilde{g}_r}{\lambda_0}\right]},$$

$$\frac{d}{dx}I_r^{\mathrm{sp}} = \frac{1}{\lambda_0}(I_r^{\mathrm{sp}} - 2I_l^{\mathrm{sp}}e^{-L/\lambda_0}) = \frac{\hbar}{2e}\frac{I}{\lambda_0}\frac{p_r\left[1+(1-p_l^2)\frac{\tilde{g}_l}{\lambda_0}\right] - 2p_le^{-L/\lambda_0}}{\left[1+(1-p_l^2)\frac{\tilde{g}_l}{\lambda_0}\right]\left[1+(1-p_r^2)\frac{\tilde{g}_r}{\lambda_0}\right]}.$$

From these boundary values of $dI^{\mathrm{sp}}/dx$ and from the relation,

$$\Delta V(p_l, p_r) = -\frac{2e}{\hbar}\left(\frac{p_l}{g_l}\tilde{g}_l\frac{d}{dx}I_l^{\mathrm{sp}} - \frac{p_r}{g_r}\tilde{g}_r\frac{d}{dx}I_r^{\mathrm{sp}}\right)$$

(24)

we obtain

$$\Delta V(p_l, p_r) = \frac{I}{g_l+g_r}\frac{\tilde{g}_l+\tilde{g}_r}{\lambda_0}\frac{p_l^2\left[1+(1-p_r^2)\frac{\tilde{g}_r}{\lambda_0}\right] + p_r^2\left[1+(1-p_l^2)\frac{\tilde{g}_l}{\lambda_0}\right] - 4p_lp_re^{-L/\lambda_0}}{\left[1+(1-p_l^2)\frac{\tilde{g}_l}{\lambda_0}\right]\left[1+(1-p_r^2)\frac{\tilde{g}_r}{\lambda_0}\right]},$$

(25)

up to the first order in $e^{-L/\lambda_0}$, from which $V_{\mathrm{mr}} \equiv \Delta V(p_l = -p_r = p) - \Delta V(p_l = p_r = p)$ can be obtained.

# B   $\Delta V$ near the half-quantum vortex deconfinement

In terms of transport, the defining characteristics of hqv's is that they contribute to both the charge and the spin terms to $\Delta V$,

$$\Delta V = \int_{-L/2}^{L/2} dx\, j_c - \frac{2e}{\hbar}\left(\frac{p_l}{g_l}\tilde{g}_l\frac{d}{dx}I_l^{\mathrm{sp}} - \frac{p_r}{g_r}\tilde{g}_r\frac{d}{dx}I_r^{\mathrm{sp}}\right).$$

The results presented in the main text shows that for any finite temperature the spin term should be in the same form as $\Delta V$ of Eq. (25). The charge term also differs from that of fqv's as the spin vorticity current is not uniform for $T < T_{\mathrm{BKT}}^{\mathrm{hqv}}$:

$$j_c \propto \frac{1}{2} \sum_\sigma \left(I + \sigma \frac{2e}{\hbar} I^{\mathrm{sp}}\right) \left|I + \sigma \frac{2e}{\hbar} I^{\mathrm{sp}}\right|^{\frac{2T_{\mathrm{BKT}}^{\mathrm{hqv}}}{T}} \approx I |I|^{\frac{2T_{\mathrm{BKT}}^{\mathrm{hqv}}}{T}} \left[1 + \frac{T_{\mathrm{BKT}}^{\mathrm{hqv}}}{T}\left(\frac{2T_{\mathrm{BKT}}^{\mathrm{hqv}}}{T} + 1\right)\left|\frac{2e}{\hbar}\frac{I^{\mathrm{sp}}}{I}\right|^2\right].$$

Altogether we obtain up to the first order in $e^{-L/\tilde{\lambda}_{\mathrm{eff}}}$

$$
\begin{aligned}
\Delta V(p_l, p_r) =& \frac{L}{w} \tilde{\rho}_0^{\mathrm{hqv}} I \left|\frac{I}{I_0^{\mathrm{hqv}}}\right|^{2T_{\mathrm{BKT}}^{\mathrm{fqv}}/T} \left[1 + \frac{\tilde{\lambda}_{\mathrm{eff}}}{2L}\frac{T_{\mathrm{BKT}}^{\mathrm{hqv}}}{T}\left(\frac{2T_{\mathrm{BKT}}^{\mathrm{hqv}}}{T} + 1\right) C(p_l, p_r)\right] \\
&+ \frac{I}{g_l + g_r} \frac{\tilde{g}_l + \tilde{g}_r}{\tilde{\lambda}_{\mathrm{eff}}} \frac{p_l^2 \left[1 + (1 - p_r^2)\frac{\tilde{g}_r}{\tilde{\lambda}_{\mathrm{eff}}}\right] + p_r^2 \left[1 + (1 - p_l^2)\frac{\tilde{g}_l}{\tilde{\lambda}_{\mathrm{eff}}}\right] - 4 p_l p_r e^{-L/\tilde{\lambda}_{\mathrm{eff}}}}{\left[1 + (1 - p_l^2)\frac{\tilde{g}_l}{\tilde{\lambda}_{\mathrm{eff}}}\right]\left[1 + (1 - p_r^2)\frac{\tilde{g}_r}{\tilde{\lambda}_{\mathrm{eff}}}\right]},
\end{aligned}
\tag{26}
$$

where $\tilde{\rho}_0^{\mathrm{hqv}}$ is a phenomenological constant in the units of 2D resistivity and the constant

$$
\begin{aligned}
C(p_l, p_r) =& \frac{p_l^2 \left[1 + (1 - p_r^2)\frac{\tilde{g}_r}{\tilde{\lambda}_{\mathrm{eff}}}\right]^2 + p_r^2 \left[1 + (1 - p_l^2)\frac{\tilde{g}_l}{\tilde{\lambda}_{\mathrm{eff}}}\right]^2 - 2 p_l p_r \left[2 + (1 - p_l^2)\frac{\tilde{g}_l}{\tilde{\lambda}_{\mathrm{eff}}} + (1 - p_r^2)\frac{\tilde{g}_r}{\tilde{\lambda}_{\mathrm{eff}}}\right] e^{-L/\tilde{\lambda}_{\mathrm{eff}}}}{\left[1 + (1 - p_l^2)\frac{\tilde{g}_l}{\tilde{\lambda}_{\mathrm{eff}}}\right]^2 \left[1 + (1 - p_r^2)\frac{\tilde{g}_r}{\tilde{\lambda}_{\mathrm{eff}}}\right]^2} \\
&+ \frac{4L}{\tilde{\lambda}_{\mathrm{eff}}} \frac{p_l p_r e^{-L/\tilde{\lambda}_{\mathrm{eff}}}}{\left[1 + (1 - p_l^2)\frac{\tilde{g}_l}{\tilde{\lambda}_{\mathrm{eff}}}\right]\left[1 + (1 - p_r^2)\frac{\tilde{g}_r}{\tilde{\lambda}_{\mathrm{eff}}}\right]}
\end{aligned}
$$

that vanishes when $p_l = p_r = 0$. From this, we obtain the magnetoresistance of Eq. (16) by taking the limit where both $L \gg \tilde{\lambda}_{\mathrm{eff}}$ and $I \ll I_0^{\mathrm{hqv}}$ are satisfied.

The charge vorticity current is uniform for $T > T_{\mathrm{BKT}}^{\mathrm{hqv}}$, in which case

$$\Delta V = \frac{L}{w}\left(\frac{\pi\hbar}{e}\right)^2 \mu_{\mathrm{hqv}} n_f I + \frac{I}{g_l + g_r}\frac{\tilde{g}_l + \tilde{g}_r}{\tilde{\lambda}_0} \frac{p_l^2 \left[1 + (1 - p_r^2)\frac{\tilde{g}_r}{\tilde{\lambda}_0}\right] + p_r^2 \left[1 + (1 - p_l^2)\frac{\tilde{g}_l}{\tilde{\lambda}_0}\right] - 4 p_l p_r e^{-L/\tilde{\lambda}_0}}{\left[1 + (1 - p_l^2)\frac{\tilde{g}_l}{\tilde{\lambda}_0}\right]\left[1 + (1 - p_r^2)\frac{\tilde{g}_r}{\tilde{\lambda}_0}\right]}
\tag{27}$$

easily reduces to Eq. (17).

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
