# Peer review of "Berezinskii-Kosterlitz-Thouless transition transport in spin-triplet superconductor"

_SciPost Physics_

## Round 2 · Referee Report · Anonymous · 2021-7-26

Report

Dear Editor,
hereby I submit my report on “Berezinskii-Kosterlitz-Thouless transition transport in spin-triplet superconductor” by Suk Bum Chung and Se Kwon Kim.

The authors study transport signatures of BKT transitions in spin-triplet superconductors. The paper is interesting and of relevance for the community working on superconductivity. For this reason, I think the manuscript is worth publishing but before I would recommend to address some concerns that will help to improve the manuscript.

Please, let me know if you have any question.

Best regards,

Jorge Cayao

Requested changes

COMMENTS FOR THE AUTHORS:

1. In the abstract, the authors should try to highlight the system they investigate. I understand it is a spin-triplet superconductor, but what type of system it is? Otherwise, it remains unclear.

2. In the first part of the introduction, the authors highlight spin-triplet superconductors. Here, I would recommend to include into the discussion other spin-triplet superconductors such as those due to spin-orbit coupling. See for instance:

Phys. Rev. Lett. 87, 037004 (2001)
Phys. Rev. B 92, 134512 (2015)
Phys. Rev. B 98, 075425 (2018)

3. In the second paragraph of the introduction, it is not clear to me what is the motivation of the authors for carrying out this study. In particular, I recommend the authors to expand the discussion on how a two component condensate in 2D, with multiple BKT transitions involving charge and spin degrees of freedom, “naturally” raises the question of robustness of spin transport.

4. In the introduction, I recommend to put in a single paragraph the text starting with “In this Letter ….”. This will help the readers to spot what is done in the manuscript. I also recommend here to briefly say what is the system the authors study.

5. In several instances of the work, I found that the authors just directly write down mathematical expressions. While at some point it is fine, I think it would help the reader if the authors include a sort of derivation or guide on how to obtain those expressions. This includes equations before (2), (2), (4), (5), (6). The authors should guide the reader.

6. The authors refer to Eq.(5) a the Josephson relation. However, most of the community is familiar with the 1t and 2nd Josephson relations involving the superconducting phase. I think the authors refer here to the Josephson energy. Please clarify.

7. In section 3, the authors use a junction setup to get signatures of each BKT transition. As mentioned before, the authors write down expressions Eqs.(7-9) but I believe they need to make it simpler for the reader. My concern is that the authors should try to make the paper self-contained and should guide the reader in the derivation of these expressions without implying to write lengthy and complicated equations.

8. In relation to the results, the authors only provide expressions. While it is nice to have expressions, I believe the authors need to make some representative plots showing their main results in subsections 3.2, 3.3, and 3.4.

9. It would be very helpful for the reader if the authors include some additional physical discussion of their results. For instance, why the authors look at the magnetoresistance voltages, etc.

  • validity: good
  • significance: good
  • originality: good
  • clarity: ok
  • formatting: good
  • grammar: good

Author:  Suk Bum Chung  on 2021-09-13

(in reply to Report 1 on 2021-07-26)

Dear Referee,

We appreciate your comments and valuable queries, which have helped us improve the clarity and quality of our manuscript. Please find detailed point-by-point responses to your questions and suggestions in the attached file. The corresponding modifications (highlighted in red) are incorporated in the revised manuscript that we are resubmitting. We believe that our manuscript has been improved sufficiently for publication in SciPost Physics.

With best regards,
Suk Bum Chung and Se Kwon Kim

Attachment:

Response_Referee1.pdf

---

## Round 2 · Referee Report · Anonymous · 2021-8-15

Report

This manuscript contains some original ideas on the interplay between different types of vortices (generated by charge and spin U(1) topological defects) and their connection to both charge and spin transport n triplet superconductors. Many ideas and concepts in this manuscript follow from previous works by one of the authors with other authors as well as the current one (see Refs. 22, 28, and 45 in the current manuscript).

I have some concerns on the claimed phase diagram though. The effective vortex action of Eq. (2) contains two integer vortex fields, namely, $m_i^c$ and $m_i^{sp}$, referring to charge and spin vortices, respectively. This effective action follows by straightforwardly integrating out the phases in the Villain action. However, these phase variables being decoupled [see, for instance, Eq. (5) in https://journals.aps.org/prl/abstract/10.1103/PhysRevLett.99.197002 , which is one of the authors of the current submission (Chung) is the first author], and therefore this leads to two BKT transition temperatures, which are determined by the charge and spin stiffnesses. The unnumbered equation after Eq. (2) for the BKT transition temperature is simply the sum of these just mentioned transition temperatures. This does not seem to naturally follow from Eq. (2). To add to the confusion, other BKT temperatures are referred to in the text ($T^{\rm dm}_{\rm BKT}$, $T^{\rm fqv}_{\rm BKT}$, and $T^{\rm hqv}_{\rm BKT}$). The authors have to clarify how these are obtained.

It might also turn out that a coupling between the charge vortices and the magnetization has to be considered at the free energy level, leading to a reformulation of Eq. (2). In fact, I would expect the term $\sim(\nabla\times\vec{A})^2$ in the Ginzburg-Landau free energy to be replaced by something $\sim(\nabla\times\vec{A}-4\pi\vec{M})^2$, where $\vec{M}$ is the magnetization. This would suggest a coupling between the charge vortex and $s_z$. Since $s_z$ is conjugated to $\alpha$, this would ultimately couple the charge and spin vortices.

I would like to ask the authors to clarify the points above before I can make a final assessment.

  • validity: -
  • significance: -
  • originality: -
  • clarity: -
  • formatting: -
  • grammar: -

Author:  Suk Bum Chung  on 2021-09-13

(in reply to Report 2 on 2021-08-15)

Dear Referee,

We appreciate your comments and valuable queries, which have helped us improve the clarity and quality of our manuscript. Please find detailed point-by-point responses to your questions and suggestions in the attached file. The corresponding modifications (highlighted in red) are incorporated in the revised manuscript that we are resubmitting. We believe that our manuscript has been improved sufficiently for publication in SciPost Physics.

With best regards,
Suk Bum Chung and Se Kwon Kim

Attachment:

Response_Referee2.pdf

---

## Editorial Decision

resubmitted